# Anti-Inflammatory Effect of the Natural H_2_S-Donor Erucin in Vascular Endothelium

**DOI:** 10.3390/ijms232415593

**Published:** 2022-12-09

**Authors:** Valerio Ciccone, Eugenia Piragine, Era Gorica, Valentina Citi, Lara Testai, Eleonora Pagnotta, Roberto Matteo, Nicola Pecchioni, Rosangela Montanaro, Lorenzo Di Cesare Mannelli, Carla Ghelardini, Vincenzo Brancaleone, Lucia Morbidelli, Vincenzo Calderone, Alma Martelli

**Affiliations:** 1Department of Life Sciences, University of Siena, Via Aldo Moro 2, 53100 Siena, Italy; 2Department of Pharmacy, University of Pisa, Via Bonanno Pisano 6, 56126 Pisa, Italy; 3Interdepartmental Research Center “Nutrafood: Nutraceutica e Alimentazione per la Salute”, University of Pisa, 56126 Pisa, Italy; 4Interdepartmental Research Center “Biology and Pathology of Ageing”, University of Pisa, 56126 Pisa, Italy; 5Research Centre for Cereal and Industrial Crops, CREA Council for Agricultural Research and Economics, Via di Corticella 133, 40134 Bologna, Italy; 6Research Centre for Cereal and Industrial Crops, CREA Council for Agricultural Research and Economics, S.S. 673 Km 25,200, 71122 Foggia, Italy; 7Department of Science, University of Basilicata, Via Ateneo Lucano 10, 85100 Potenza, Italy; 8Pharmacology and Toxicology Section, Department of Neuroscience, Psychology, Drug Research and Child Health (NEUROFARBA), University of Florence, Viale Gaetano Pieraccini, 6, 50139 Florence, Italy; 9Interuniversity Center for Studies on Bioinspired Agro-Environmental Technology (BAT Center), 80055 Naples, Italy

**Keywords:** vascular inflammation, endothelial dysfunction, hydrogen sulfide, erucin, isothiocyanates, *Eruca sativa* Mill.

## Abstract

Vascular inflammation (VI) represents a pathological condition that progressively affects the integrity and functionality of the vascular wall, thus leading to endothelial dysfunction and the onset of several cardiovascular diseases. Therefore, the research of novel compounds able to prevent VI represents a compelling need. In this study, we tested erucin, the natural isothiocyanate H_2_S-donor derived from *Eruca sativa* Mill. (*Brassicaceae*), in an in vivo mouse model of lipopolysaccharide (LPS)-induced peritonitis, where it significantly reduced the amount of emigrated CD11b positive neutrophils. We then evaluated the anti-inflammatory effects of erucin in LPS-challenged human umbilical vein endothelial cells (HUVECs). The pre-incubation of erucin, before LPS treatment (1, 6, 24 h), significantly preserved cell viability and prevented the increase of reactive oxygen species (ROS) and tumor necrosis factor alpha (TNF-α) levels. Moreover, erucin downregulated endothelial hyperpermeability and reduced the loss of vascular endothelial (VE)-Cadherin levels. In addition, erucin decreased vascular cell adhesion molecule 1 (VCAM-1), cyclooxygenase-2 (COX-2) and microsomal prostaglandin E-synthase 1 (mPGES-1) expression. Of note, erucin induced eNOS phosphorylation and counteracted LPS-mediated NF-κB nuclear translocation, an effect that was partially abolished in the presence of the eNOS inhibitor L-NAME. Therefore, erucin can control endothelial function through biochemical and genomic positive effects against VI.

## 1. Introduction

Vascular inflammation is a pathological condition characterized by an acute or chronic pro-inflammatory stimulus which affects the integrity and the functionality of the vascular wall. The first target of inflammation at the vascular level is represented by the endothelial layer, which consequently becomes dysfunctional and more permeable. Endothelial dysfunction leads, in turn, to decreased nitric oxide (NO) biosynthesis and loss of vascular tone control, and it is prodromic of several cardiovascular (CV) diseases such as hypertension, thromboembolism and the onset of the atherosclerotic plaques. On the other hand, the increase in endothelial permeability leads to extravasation, leucocyte adhesion and migration, and the spread of inflammation [1]. The protection of the vascular wall against chronic and acute pro-inflammatory stimuli is still an unmet medical need, as demonstrated also during the last pandemic in which the cytokine storm induced by SARS-CoV-2 affected the endothelium leading to severe CV events [2]. In this scenario, the research of compounds able to protect the vascular tree, and in particular the endothelial tissue, against inflammation, represents a challenge for pharmacologists. Several synthetic and natural compounds have been proposed on the basis of their ability to inhibit the release of one or more cytokines or the expression/activity of some transcriptional factors. However, these results are often derived from studies which did not focus on the reduction or the prevention of the vascular inflammation as primary outcome [3,4]. In recent years, some evidence showed that the endogenous gasotransmitter hydrogen sulfide (H_2_S) plays a role in the control of inflammation at a vascular level, suggesting a potential use of exogenous molecules able to release H_2_S (H_2_S-donors) in the management of vascular inflammation and consequent endothelial dysfunction [5,6,7,8]. Among the large number of H_2_S-donors, the natural ones demonstrated in several studies the profile of “smart” H_2_S-donors, characterized by a slow and gradual release of H_2_S. Such a “smart” release is often dependent on the presence of organic thiols, a behavior that seems more suitable for pharmacological purposes [9,10,11,12,13,14,15]. In recent years, erucin, the isothiocyanate H_2_S-donor derived from *Eruca sativa* Mill. (the common rocket or arugula), has been investigated for its CV properties linked to the ability to release H_2_S. In recent studies, erucin demonstrated vasorelaxing and anti-hypertensive properties in in vitro and in vivo experimental models. Moreover, it also exhibited the ability to protect human endothelial and aortic smooth muscle cells against the damage induced by high glucose [16,17]. However, the latter pro-inflammatory stimulus does not fully reproduce a sudden inflammatory condition characterized by a massive cytokine storm, thus limiting a wide and complete understanding of the pharmacological potentiality of erucin in the prevention of vascular inflammation induced by multiple stimuli. On this basis, we here aimed to unveil the possible anti-inflammatory effect of the natural H_2_S-donor erucin in an in vivo model of acute peritonitis, also addressing the possible molecular pathways involved by using human cultured endothelial cells. In particular, endothelial cells were pretreated with erucin before the exposure to *E. coli* lipopolysaccharide (LPS), evaluating the endothelial functional and molecular responses at times covering the onset of the acute inflammatory response occurring in the time range up to 24 h [18,19,20]. 

## 2. Results

### 2.1. In Vivo Model of Peritonitis

The anti-inflammatory effect of erucin was first evaluated in vivo by using a peritonitis model induced by LPS injection (0.1 mg/kg, 100 µL i.p.). Peritoneal lavage highlighted that LPS administration induced an increase of neutrophil transmigration in the peritoneal cavity, as shown by CD11b/Ly6G double positive cells (Figure 1A), where Ly6G represents a specific marker expressed by murine neutrophils and CD11b is a crucial integrin involved in the transmigration process. In particular, LPS administration resulted in about 40% of neutrophils compared to 4% displayed in control (Veh). The pretreatment with erucin (1 µmol/kg, 100 µL i.p.) was able to impair the inflammatory response in mice treated with LPS, reducing the neutrophil transmigration by 60% (Figure 1A,B).

### 2.2. H_2_S-Releasing Properties of Erucin via Amperometric Assay

The assessment of the H_2_S release was achieved by using H_2_S-selective minielectrode and incubating erucin 1 mM in the presence or in absence of L-Cysteine. As shown in Figure 2, the H_2_S formation was almost negligible in the absence of L-Cysteine, whereas in the presence of L-Cysteine 4 mM a growing generation of H_2_S was detected, reaching a 2 µM-stable steady state after 20 min. The L-Cysteine-dependent H_2_S release makes erucin a “smart” H_2_S-donor, able to generate the gaseous molecule only when in contact with biological substrates. Indeed, the thiol group of the L-Cysteine is able to react with the NCS moiety leading to the formation of H_2_S, as reported by Lin et al. [21].

### 2.3. Effect of Erucin on Endothelial Permeability and VE-Cadherin Changes in Response to LPS 

Inflammatory injury of the endothelium leads to endothelial dysfunction [22]. In order to investigate the contribution of erucin for endothelium integrity, a permeability assay was performed on confluent HUVEC monolayers exposed to LPS for short times. In time course experiments, an increase of endothelial permeability was observed after LPS stimulation (1 µg/mL) from 30 min to 180 min of exposure (Figure A1). At 240 min, HUVECs recovered the damage. Subsequent experiments were performed at 60 min of LPS stimulation. LPS significantly increased endothelial permeability compared with the control condition. The pre-incubation of endothelial monolayers with 3 µM erucin prevented LPS-induced hyperpermeability and brought it back to the basal level, showing a protective effect per se (Figure 3A). To further strengthen our findings, the immunofluorescence analysis of tight junction proteins in endothelial cells was also performed. VE-Cadherin was evaluated as a representative marker of endothelial tight junctions. In the basal control condition, HUVECs confluent monolayer expressed VE-Cadherin with a plasmalemmal localization at cell–cell contacts, as well as cytoplasmatic and perinuclear accumulation. Exposure to LPS caused the fading of fluorescence intensity, which was partially reverted by pre-treatment with erucin (3 µM) (Figure 3B,C). Collectively, these data indicate that erucin maintains the endothelial integrity and prevents the acute injury induced by LPS on the barrier function of the endothelium.

### 2.4. Preventive Effects of Erucin against Cell Viability Reduction and Intracellular ROS Increase in HUVECs Challenged with LPS 

Endothelial cells were exposed to LPS for longer times of incubation (6 and 24 h), which are compatible with biochemical, molecular and functional responses related to inflammation (6 h) and mimic a persistent and progressive inflammatory state (24 h). The treatment of HUVECs with LPS 1 µg/mL significantly decreased cell viability (% of cell viability vs. vehicle: 78.46 ± 1.28 after 6 h and 83.49 ± 2.80 after 24 h). 

Only the pre-incubation of the highest concentration of erucin (3 µM) led to a significant, although modest, prevention of such decrease (% cell viability vs. vehicle was 82.79 ± 2.62 for 0.3 µM, 81.75 ± 2.56 for 1 µM and 89.54 ± 3.18 for 3 µM after 6 h, and 78.45 ± 2.65 for 0.3 µM, 85.61 ± 3.30 for 1 µM and 94.10 ± 2.29 for 3 µM after 24 h) (Figure 4A,B). In addition, LPS induced a significant increase in intracellular ROS production (% ROS vs. vehicle: 120.42 ± 3.32 after 6 h and 114.28 ± 2.64 after 24 h). The pre-incubation of increasing concentrations of erucin prevented the increase of LPS-induced ROS levels in a concentration-dependent manner (% cell viability vs. vehicle was 107.50 ± 5.09 for 0.3 µM, 105.83 ± 4.59 for 1 µM and 100.67 ± 1.96 for 3 µM after 6 h, and 110.88 ± 4.17 for 0.3 µM, 104.60 ± 4.97 for 1 µM and 93.60 ± 4.16 for 3 µM after 24 h) (Figure 4C,D).

Of note, the preventive effect exhibited by erucin against endothelial damage after incubation of LPS for different times (6 h and 24 h) was quite superimposable, indicating that the initial protection of the endothelium promoted by erucin is maintained even after a prolonged exposure to the pro-inflammatory stimulus.

### 2.5. Evaluation of Erucin Effect on VCAM-1, E-Selectin, COX-2 and mPGES-1 Expression Induced by LPS 

The biochemical mechanisms associated with the protective effect of erucin on the endothelium were then investigated. Key markers and enzymes involved in inflammation in response to LPS were analyzed and the influence of erucin was investigated. Assessing the genetic modulation of inflammatory genes, we found a significant increased expression of inducible mPGES-1 transcript in HUVECs treated with LPS, which was impaired by erucin pre-treatment (Figure 5A). The regulation of mPGES-1 expression was also observed at the protein level (Figure 5B). Erucin’s pre-incubation prevented the overexpression of endothelial adhesion molecule VCAM-1, and inflammatory mediators COX-2 and mPGES-1, induced by LPS (Figure 5B,C). Analyzing the markers of cell-cell interaction and cellular mobilization, we observed a plasmalemmal localization of VCAM-1 in HUVECs treated with LPS, and a marked reduction of its intensity following erucin pre-treatment (Figure 5D,E). E-selectin localization documented no relevant differences in relation with treatments (Figure A2). Together the data suggest that erucin pre-treatment exerts a protective effect in endothelial cells exposed to LPS injury.

### 2.6. Erucin Prevents the Increase in TNF-α Levels, Expression of Inflammatory Markers and Enzymes (iNOS, COX-2/mPGES-1 and VCAM-1), and the Number of NO-Positive Cells Induced by Treatment with LPS for 24 h

The preventive effects of erucin against LPS-induced increase in the inflammatory marker TNF-α were then investigated. Treatment of HUVECs with LPS (1 µg/mL) for 24 h, but not 6 h, led to a significant increase in TNF-α production. Indeed, TNF-α levels measured in the supernatant of vehicle-treated cells were 0.11 ± 0.10 pg/mL, while TNF-α levels detected in the supernatant of LPS-exposed cells were much higher (2.11 ± 0.31 pg/mL). Pre-incubation of erucin 3 µM for 1 h significantly prevented the TNF-α increase induced by LPS (TNF-α levels: 0.38 ± 0.09 pg/mL) (Figure 6A). At 24 h, the impairment of LPS-mediated induction of mPGES-1 transcription by erucin seen at 6 h (Figure 5A) was exhausted (Figure 6B), suggesting an early modulation of endothelial transcription by erucin. At the same time, the protein expression of iNOS was drastically augmented in cells treated with LPS compared with vehicle, while pre-treatment with erucin (3 µM) reduced its expression to a basal level (Figure 6C,D). Similar results were obtained on markers and enzymes representing key inflammatory pathways. Indeed, the LPS-mediated COX-2, mPGES-1 and VCAM-1 protein increase was abolished in the presence of erucin (Figure 6C,D). With the increase in iNOS expression, the number of NO-positive HUVECs (both total and live) challenged with LPS 1 µg/mL for 24 h was significantly higher than the number of NO-positive cells treated with vehicle, suggesting a possible role for iNOS in the vascular damage induced by LPS. Pre-treatment with erucin 3 µM significantly prevented the increase in NO-positive cells following the incubation of LPS (Figure 6E,F).

### 2.7. Erucin Regulates the Nuclear Translocation of NF-kB Induced by LPS through eNOS Activation

In order to gain insights into endothelial mediated effects, eNOS activation was assessed in HUVECs exposed to erucin. A significant activation, evaluated as eNOS phosphorylation at Ser1177, was observed at short times (5–10 min) of treatment, comparable with that induced by the vascular endothelial growth factor (VEGF) (Figure 7A–C). As eNOS-derived NO prevents vascular inflammation, the nuclear translocation of nuclear factor kappa-light-chain-enhancer of activated B cells (NF-kB) p65 subunit, a transcription factor which orchestrates gene expression program mediated by LPS [23], was assessed in the presence of the eNOS inhibitor N-nitro-L-arginine methylester (L-NAME) (200 µM, 30 min). After 2 h, LPS induced NF-kB p65 subunit nuclear localization, a prerequisite for transcription, which was reduced by erucin pre-treatment (Figure 7D). Interestingly, L-NAME pre-incubation partially abolished the erucin activity, as an increase of NF- kB p65 subunit nuclear localization was observed (Figure 7D). These results showed how erucin counteracts LPS-induced gene transcription. It promotes fast eNOS activation, which in turn reduces NF-kB p65 subunit nuclear translocation.

## 3. Discussion

Vascular inflammation is a degenerative process that results from acute or chronic exposure of the endothelium to pro-inflammatory and oxidative stimuli. It is prodromic for the development of hypertension, atherosclerosis, and other CV disorders [24], also representing a major trigger for non-CV diseases. In recent years, the emerging antioxidant and anti-inflammatory role of H_2_S supported a possible use of H_2_S-donors in the prevention and treatment of vascular inflammation and related disorders [25]. In this regard, we recently described the H_2_S-releasing properties of the natural isothiocyanate erucin in vascular cells [16,17], as well as its promising protective effects against high-glucose induced vascular inflammation [17]. In this work, using a cell-free amperometric technique, we first confirmed the thiol-dependent H_2_S-releasing properties of erucin, previously demonstrated in a cell-based assay on HUVECs [17]. We then proceeded with a further characterization of the potential anti-inflammatory activity of this natural “smart” H_2_S-donor in the vasculature. 

It is widely known that H_2_S displays anti-inflammatory effects in vivo, where it has been shown to specifically limit the leukocytes trafficking that drives inflammation [26,27]. Indeed, we carried out an in vivo model to test the efficacy of erucin as an “H_2_S-releasing” anti-inflammatory drug. Interestingly, erucin here shows a potent inhibition of activated neutrophils identified as CD11b/Ly6G positive leukocytes. In fact, the acute administration of erucin significantly prevented the increase of neutrophil transmigration in the mouse peritoneum, confirming the role of a smart H_2_S-donor exerted by erucin, as well as the observations on H_2_S biology in inflammation [28].

Following this evidence, we wanted to investigate whether the changes in the key process of inflammatory reactions could be targeted by erucin. One of the main features of vascular inflammation is represented by the modification of endothelial permeability. Indeed, an inflammatory stimulus could trigger a condition of hyperpermeability, which progressively leads to vascular damage, loss of the barrier function of the endothelium, widespread inflammatory mediators and multiorgan dysfunction [29]. Therefore, preventing endothelial hyperpermeability represents a potential pharmacological strategy to counteract the deleterious consequences of vascular inflammation. In this paper, we demonstrated that the exposure of endothelial cells to the pro-inflammatory stimulus LPS led to a significant increase in endothelial permeability, which was accompanied by a reduced expression of the vascular adhesion molecule VE-Cadherin. Pre-incubation with erucin prevented endothelial hyperpermeability and almost preserved basal levels of VE-Cadherin, suggesting that this H_2_S-donor might contribute to the maintenance of vascular integrity and functionality under inflammatory conditions, as also assessed in a high glucose environment [17]. 

The existence of a vicious circle between inflammation and persistent oxidative stress has been widely described in multiple vascular disorders [30,31]. Also in our work, endothelial cells challenged with LPS exhibited enhanced intracellular ROS production and reduced cell viability. All these detrimental events were prevented by pre-incubation with the H_2_S-donor erucin, further supporting its protective effects against vascular inflammation and oxidative stress.

It is worthy of note that the exposure of endothelial cells to LPS also induced the release of the pro-inflammatory cytokine TNF-α, which was accompanied by an increase in the levels of both COX-2 and microsomal enzyme mPGES-1, a downstream target of COX-2 that contributes to prostaglandin E2 (PGE2) production [32]. This finding confirms the pivotal role of inflammation in the vascular damage induced by the prolonged exposure of endothelial cells to infective agent degradation products such as LPS. The pre-incubation of erucin attenuated the inflammatory response by reducing TNF-α levels, as well as COX-2 and mPGES-1 protein expression. Of note, erucin also prevented the increase in mPGES-1 mRNA levels induced by LPS, indicating that the natural H_2_S-donor exerts anti-inflammatory properties not only via post-translational modifications of proteins but also by the transcriptional modulation of specific inflammatory genes. Moreover, the protective effect of erucin pre-treatment for 1 h was evident after both short (1 h) and long times (24 h) of LPS exposure, documenting the early genomic and biochemical mechanisms responsible for the functional properties on the vascular endothelium.

The pro-inflammatory stimulus LPS induced an increase of other typical markers of inflammation, iNOS and the gasotransmitter NO. Erucin significantly prevented the increase in both iNOS expression and long term NO levels in endothelial cells, further confirming its potential use as an anti-inflammatory agent in the prevention of vascular inflammation and related disorders. 

Finally, the prolonged exposure of endothelial cells to LPS led to an increase in intracellular levels of the adhesion molecule VCAM-1 and promoted its plasmalemmal localization. On the contrary, no differences have been observed in E-selectin localization. Erucin also exhibited anti-inflammatory effects by lowering VCAM-1 levels and preventing VCAM-1 plasmalemmal localization. 

It is interesting to note that the transcription of most inflammatory mediators (i.e., COX-2, mPGES-1, TNF-α and iNOS) is positively modulated by the pro-inflammatory transcription factor NF-κB. For instance, the treatment of macrophages with LPS increased the expression of COX-2 and mPGES-1, probably through the activation of NF-κB [33]. Moreover, LPS enhanced NF-κB levels in cultured neurons, thus increasing iNOS derived NO production [34]. In this regard, we recently demonstrated that the H_2_S-donor erucin reduced the expression of NF-κB in endothelial cells challenged with a pro-inflammatory stimulus [17], suggesting that this key mechanism explains, at least in part, the anti-inflammatory effects of erucin in the endothelium. This finding is here strengthened by the ability of erucin to inhibit LPS-mediated NF-κB nuclear translocation. This hypothesis is partially confirmed by the work by Cho and colleagues, who reported an inhibitory effect of erucin against the LPS-induced activation of NF-κB signaling in macrophages [35] and, in general, by the inhibitory effect of H_2_S on NF-κB activation in the vascular endothelium [7].

Given the regulation of fast activation of mPGES-1 transcription, we focused on early mechanisms potentially activated by erucin. After 5–10 min of stimulation, erucin was able to ignite eNOS phosphorylation at Ser1177 in a manner similar to the receptor linked phospho-eNOS due to VEGF. To correlate eNOS activation to the anti-inflammatory effects of erucin, endothelial cells were treated with the eNOS inhibitor L-NAME, and NF-κB localization was assessed following a time related to transcription factor activity (2 h). Erucin’s ability to inhibit NF-κB nuclear redistribution was blunted by eNOS inhibition, documenting also that the endogenous gasotransmitter NO participates in the protective effect of erucin. On the whole, erucin promoted endothelial-derived protective NO due to fast eNOS activation, while in the long run it inhibits iNOS-derived NO, known to be a player of vascular inflammation and endothelial dysfunction [36,37]. An overlapping pathway has been demonstrated to be ignited by the H_2_S donor/generator zofenoprilat, which exerts a protective role in vascular endothelial functions impaired by inflammatory stimuli, such as interleukin 1β [38].

As a main limitation of our study, the use of LPS as pro-inflammatory stimulus does not allow for the reproduction of the chronic, weaky vascular inflammatory state typical of many CV and also non-CV diseases. On the contrary, LPS induces an acute and massive cytokine storm, which is characteristic of many bacterial and viral infections. If not resolved, such an inflammatory cascade can lead to serious consequences in multiple organs and tissues. Hence, this model cannot be used to induce chronic vascular inflammation in vivo, not allowing the investigation of the preventive effects of erucin in the long period. 

On the basis of our results, future experiments will be aimed at evaluating the endothelial protection promoted by erucin on isolated vessels (i.e., by means of histological and functional assays) in the acute model of LPS-induced vascular inflammation in vivo. Furthermore, the potential preventive effect of erucin against cell damage in human vascular smooth muscle cells exposed to LPS could represent an additional aspect to be investigated in the future in order to provide a complete overview of the effects of the natural H_2_S-donor in the vasculature. As concerns the safety profile, a very recent study on acute toxicity in rats aimed at evaluating the risk of a single but high oral dose of erucin showed that the mean LD50 was 500 mg/kg [39]. Both sub-acute (28 days) and sub-chronic (90 days) oral administration of 40 mg/kg erucin revealed a moderate liver toxicity and a significant alteration of biochemical parameters, especially in female rats. On the contrary, 2.5 and 10 mg/kg erucin were safe and well-tolerated, without any evidence of toxicity in the long period [39].

In conclusion, we demonstrated, for the first time (to the best of our knowledge), that erucin displays an anti-inflammatory effect in an LPS-induced inflammatory model and protects the endothelium against vascular inflammation. These actions are based on the modulation of multiple pharmacological targets, including oxidative stress and inflammatory regulators. Together with the assumption that edible plants belonging to the *Brassicaceae* botanical family are endowed with safety and tolerability, as they have been used in the human nutrition for a long time, our results further support the translational potential of erucin in clinical studies aimed at evaluating its potential applications in the prevention of vascular inflammation and related disorders.

## 4. Materials and Methods

### 4.1. Materials

Erucin was obtained from the myrosinase-catalyzed hydrolysis of glucoerucin, isolated from *Eruca sativa* Mill. defatted seed meal and identified by GC-MS analysis as previously described [40]. Before starting the experimental procedures, erucin was dissolved in dimethyl sulfoxide (DMSO; Merck KGaA, Darmstadt, Germany) and diluted in the appropriate culture medium. Lipopolysaccharide (LPS; Merck KGaA, Darmstadt, Germany) was diluted in an aqueous solution and stored at −20 °C. Before each experiment, LPS was diluted in a culture medium up to a final concentration of 1 µg/mL. CelLytic™ MT Cell Lysis Reagent, Fluoromount Aqueous Mounting Medium, N-nitro-L-arginine methylester (L-NAME) (eNOS inhibitor) and 3 kDa FITC-Dextran were obtained from Merck KGaA, Darmstadt, Germany. VEGF (25 ng/mL) was obtained from R&D Systems, Minneapolis, MN, USA. Anti-vascular endothelial (VE)-Cadherin was obtained from Cell Signaling Technology, Danvers, MA, USA. Anti-inducible nitric oxide synthase (iNOS) was obtained from Santa Cruz Biotechnology, Dallas, TX, USA. Anti-vascular cell adhesion molecule 1 (VCAM-1) and anti-E-selectin were obtained from OriGene Technologies, Rockville, MD, USA. Anti- cyclooxygenase-2 (COX-2) and anti-microsomal prostaglandin E-synthase 1 (mPGES-1) came from Cayman Chemical, USA. Anti-β-actin, anti-peNOS Ser1177, anti-eNOS and DAPI were obtained from Merck KGaA, Darmstadt, Germany. FITC-anti-CD11b and APC-anti-Ly6G came from eBioscience, Thermo Fisher Scientific, Waltham, MA, USA. The phosphate buffered saline (PBS) buffer (Merck KGaA, Darmstadt, Germany) was adjusted to pH 7.4 at room temperature, and L-Cysteine was obtained from Merck KGaA, Darmstadt, Germany) and the NaHS is from Cayman Chemical, Ann Arbor, MI, USA.

### 4.2. Animals and LPS-Induced Peritonitis

Male C57BL/6 mice, weighing 20 to 25 g, were purchased from Charles River, Lecco, Italy, and maintained on a standard chow pellet diet with tap water supplied ad libitum. Animals were kept in a 12 h light/dark cycle and housed for a week before carrying out experiments. All experiments were performed following ARRIVE guidelines [41,42] EU recommendations (Directive 2010/63/EU) for experimental design and analysis in pharmacology care, and were authorized by the Ministero della Salute (Directive 26/2014; prot.n. 290-2018-PR). Peritonitis was induced by the intraperitoneal injection of E. coli LPS (0.1 mg/kg, 0111:B4), alone or in combination with erucin (1 µmol/kg) for 4 h. The control group (five mice) received the vehicle only (sterile PBS, 100 µL); the LPS group (five mice) received the vehicle (sterile PBS, 100 µL) 30 min before LPS injection; the LPS+erucin group (five mice) received erucin 30 min before LPS injection. After 4 h, mice were sacrificed in a CO_2_ environment and peritoneal cells were collected by using 2 mL of ice-cold sterile PBS, centrifuged, and used for flow cytometry analysis.

### 4.3. Flow Cytometry

A flow cytometry analysis on peritoneal cells was performed to quantify the extent of neutrophil recruitment in peritoneal lavage as a marker of inflammation. APC anti-Ly6G (1:200 dilution) was used to identify neutrophils while FITC-anti-CD11b (1:200 dilution) expression on Ly6G positive cells was used as a marker of cell activation. Following staining with anti-Ly6G and anti-CD11b (clone CBRM1/5; eBioscience, Thermo Fisher Scientific, Waltham, MA, USA), the cell pellets were washed and fixed with 4% PFA before performing sample analysis with a E6Bricyte flow cytometer (Mindray North America, Mahwah, NJ, USA), acquiring >10,000 events. The results were reported as dot plots displaying CD11b/Ly6G double-positive events.

### 4.4. Amperometric Evaluation of the H_2_S-Releasing Properties of Erucin

The H_2_S-releasing properties of erucin have been evaluated by an amperometric approach, through an Apollo-4000 Free Radical Analyzer (World Precision Instruments -WPI, Sarasota, FL, USA) detector and H_2_S-selective electrodes. The experiments were carried out at room temperature in a PBS buffer (Merck KGaA, Darmstadt, Germany) adjusted to pH 7.4. The H_2_S-selective electrode was equilibrated in 2 mL of the PBS buffer after reaching a stable baseline. Next, 20 μL of a DMSO solution of erucin was added at the final concentration of 1 mM. The generation of H_2_S was monitored for 20 min. When required by the experimental protocol, an excess of 4 mm of L-Cysteine (Merck KGaA, Darmstadt, Germany) was added for the mimicking of the endogenous presence of organic thiols. The relationship between the amperometric currents (recorded in pA) and the corresponding concentrations of H_2_S generated by the compound was determined by using NaHS (Cayman Chemical, Ann Arbor, MI, USA) 1 μM at pH 4.0.

### 4.5. Cell Cultures

Human umbilical vein endothelial cells (HUVECs; Thermo Fisher Scientific, Waltham, MA, USA) were cultured at 37 °C in a CO_2_ (5%) incubator in T-75 flasks. 90% confluent cells between passages 5 and 15 were used for the experiments. The culture medium was composed of basal medium (Medium 131; Thermo Fisher Scientific, Waltham, MA, USA) supplemented with an aqueous solution of L-glutamine (1%), antibiotics (100 µg/mL streptomycin and 100 U/mL penicillin), fetal bovine serum (FBS, 10%), sodium heparin (10 U/mL), basic fibroblast growth factor (5 ng/mL) and epidermal growth factor (10 ng/mL). Supplements were purchased from Merck KGaA, Darmstadt, Germany. Cell culture dishes and wells were coated with a filtered solution of gelatin from porcine skin (1%; Merck KGaA, Darmstadt, Germany), to allow cell attachment. The control of mycoplasma was routinely performed, starting from frozen vials.

### 4.6. Cell Viability

HUVECs (20,000 cells/well) were seeded in 96-well transparent plates. The following day, the culture medium was replaced with culture medium containing erucin (0.3, 1 and 3 µM) or its vehicle (0.03% DMSO). After 1 h, a freshly prepared solution of LPS (final concentration: 1 µg/mL) was co-incubated with erucin or DMSO for 6 h or 24 h. Then, cell viability was assessed using the water-soluble tetrazolium salt-1 (WST-1; Roche, Basilea, Switzerland), incubated at 37 °C in a CO_2_ incubator for 1 h. Cell viability was measured at λ = 495 nm with the microplate spectrophotometer EnSpire (PerkinElmer, Waltham, MA, USA). 

### 4.7. Measurement of Intracellular Reactive Oxygen Species (ROS) Levels

HUVECs (30,000 cells/well) were seeded in 96-well black plate wells. The day after seeding, the culture medium was replaced with fresh medium, and erucin (0.3, 1 and 3 µM) or its vehicle (0.03% DMSO) were incubated for 1 h. Next, a freshly prepared solution of LPS (final concentration: 1 µg/mL) was co-incubated with erucin or DMSO for 6 or 24 h. At the end of the treatment, intracellular ROS production was measured using the fluorescent probe 2′,7′-dichlorofluorescin diacetate (DCFDA; Merck KGaA, Darmstadt, Germany). Cells were incubated with DCFDA (20 µM in each well) in the dark for 45 min at 37 °C in a CO_2_ (5%) incubator, as previously reported [15]. Fluorescence was detected at λex = 500 nm and λem = 530 nm with an EnSpire microplate reader (PerkinElmer, Waltham, MA, USA).

### 4.8. Endothelial Permeability

HUVECs (8 × 10^4^/insert) were seeded on gelatin-coated insert membranes (Corning, NY, USA) with 0.4 µm-diameter pores and grown in 12 multiwell plates for 72 h. Confluent monolayers were pre-treated with erucin (3 µM, 1 h), and then LPS 1 µg/mL was added where required. Throughout the treatments, the cells were maintained in a medium with 10% FBS. FITC-Dextran (3 kDa, 10 μM) was used as a fluorescent probe of permeability. Every 15 min, fluorescence in the lower compartment was measured (485 and 535 nm excitation and emission, respectively) by using a microplate reader (Infinite 200 Pro, SpectraFluor, Tecan, Männedorf, Switzerland) [17]. Data are reported as fluorescence intensity.

### 4.9. Western Blot

Subconfluent endothelial cells were seeded in 60 mm Petri dishes. After 24 h, cells were pre-treated with erucin (3 µM, 1 h) and then LPS 1 µg/mL was added in medium containing 10% FBS for either 6 h or 24 h. For peNOS Ser1177 activation analysis, HUVECs were previously starved for 4 h and then treated with erucin (3 µM) or VEGF (25 ng/mL) for 5 and 10 min. At the end of stimulation, proteins were extracted, and western blots were performed as previously described [43]. Immunoblots were analyzed by densitometry using Image J 1.48v software (U.S. National Institutes of Health, Bethesda, MD, USA), and the results, expressed as arbitrary density units (A.D.U.) ± SD, were normalized toward β-actin.

### 4.10. Immunofluorescence Analysis

The cell–cell contact protein VE-Cadherin, the cell adhesion molecules VCAM-1 and E-selectin and NF-kB p65 subunit were visualized by immunofluorescence analysis. To this end, 5 × 10^4^ HUVECs were seeded on 1 cm circular glass coverslips. After 24 h, the confluent cells were washed and pre-treated with erucin (3 µM, 1 h) and then with LPS 1 µg/mL. For NF-κB p65 subunit localization, cells were previously starved for 4 h. Where appropriate, HUVECs were pre-treated with L-NAME (200 µM) for 30 min, followed by erucin (3 µM, 1 h) and then with LPS treatment 1 µg/mL for 2 h. DAPI was used to counterstain cell nuclei. An immunofluorescence analysis was performed as previously indicated on images taken using a fluorescence microscope (Nikon Eclipse TE300) [17]. 

### 4.11. RNA Isolation and Reverse Transcription-Quantitative (RT-q) PCR

Total RNA was prepared using a RNeasy Plus kit (cat. no. 74134 Qiagen GmbH, Hilden, Germany) following the manufacturer’s instructions. The quality and quantity of the purified RNA were determined by measuring the absorbance at 260/280 nm (A260/A280) using Infinite F200 Pro (Tecan, Männedorf, Switzerland). A total of 1 µg of RNA was reverse transcribed using a QuantiTect Reverse Transcription kit (cat. no. 205313 Qiagen GmbH, Hilden, Germany). RT-qPcR was performed using QuantiNova SYBR Green PcR kit (cat. no. 208056 Qiagen GmbH, Hilden, Germany) in a RotorGene qPcR machine (Qiagen GmbH, Hilden, Germany). Fold change expression was determined by the comparative ct method (Δct) normalized to 60S ribosomal protein L19 (RPL19) expression. RT-qPcR data were represented as ct (cycle threshold) value or fold increase relative to untreated cells (Vehicle) assigned to 1 [44]. The primer sequences were: mPGES-1 5′-GCTGCTGGTCATCAAGATGT-3′ and reverse 5′-CCCAGGAAGAAGACGAGAAA-3′. RPL19 forward 5′-GATGCCGGAAAAACACCTTG-3′ and reverse 5′-TGGCTGTACCCTTCCGCTT-3′. All primers were from Merck KGaA, Darmstadt, Germany.

### 4.12. ELISA Assay for TNF-α

HUVECs (200,000 cells/well) were seeded in a 6-well transparent plate. The following day, the culture medium was replaced with a culture medium containing erucin (3 µM) or its vehicle (0.03% DMSO). After 1 h, a freshly prepared solution of LPS (final concentration: 1 µg/mL) was co-incubated with erucin or DMSO for either 6 h or 24 h. At the end of treatment, an ELISA assay for tumor necrosis factor-α (TNF-α; Thermo Fisher Scientific, Waltham, MA, USA) was performed in cell supernatants, according to the manufacturer’s instructions. Absorbance was measured at 450 nm using the microplate reader EnSpire (PerkinElmer, Waltham, MA, USA).

### 4.13. Assay to Measure NO Levels

HUVECs (200,000 cells/well) were seeded in a 6-well transparent plate. Then, cells were treated as in Section 4.12 At the end of treatment (24 h), cells were gently removed with a scraper and centrifuged at 1100 rpm for 5 min. Supernatants were discarded, cell pellets were resuspended in the assay buffer (5 × 10^6^ cells/mL), and the percentage of NO-positive cells was detected using the Muse^®^ Nitric Oxide Kit (Merck KGaA, Darmstadt, Germany). The kit has been designed to measure NO levels using a membrane-permeable reagent (DAX-J2™ Orange) that generates fluorescence after oxidation by NO. Moreover, the dead cell marker (7-AAD) allows for the distinguishing among NO-positive live and dead cells. The NO assay was performed according to the manufacturer’s instructions. Briefly, cell suspensions (20 µL each) were incubated with the Muse^®^ Nitric Oxide Reagent working solution (100 µL) for 30 min in the 37 °C incubator with 5% CO_2_. Then, 90 µL of Muse 7-AAD working solution was added in the dark. After an incubation period of 5 min, analysis was performed using the Muse^®^ Cell Analyzer (Merck KGaA, Darmstadt, Germany). 

### 4.14. Data Analysis

The results are shown as mean ± SEM or mean ± SD. A statistical analysis was performed with the software GraphPad Prism 5.0, San Diego, CA, USA. A one-way ANOVA followed by Bonferroni’s post hoc test was selected for statistical analysis. *p*-value < 0.05 indicates statistical significance. Results were derived from at least three independent experiments, each performed in triplicate.

## 5. Conclusions

The vascular endothelium is the first layer to be exposed to many dangerous oxidative and pro-inflammatory stimuli circulating in the bloodstream, which progressively leads to vascular hyperpermeability and endothelial dysfunction. Moreover, when the barrier function of the endothelium is compromised, reactive oxygen species and inflammatory mediators can pass through the disrupted endothelial layer and damage vascular smooth muscle cells, as well as multiple organs and tissues. Therefore, the preservation of vascular integrity and functionality represents a main challenge for pharmacological and nutraceutical purposes. Our results confirm that the H_2_S-donor erucin exhibits antioxidant and anti-inflammatory effects in endothelial cells exposed to detrimental pro-inflammatory stimuli, also preventing endothelial hyperpermeability. Moreover, the acute administration of erucin counteracts the increase of neutrophil transmigration in vivo. Therefore, erucin is a natural H_2_S-donor of great pharmacological and nutraceutical interest, with potential future applications in the prevention of vascular inflammation and related diseases.

## Figures and Tables

**Figure 1 ijms-23-15593-f001:**
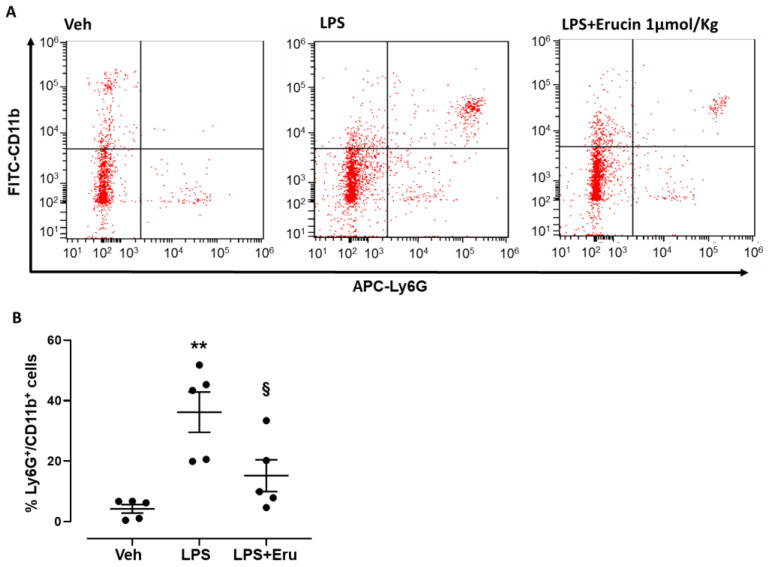
Preventive effects of erucin against LPS-induced neutrophil transmigration in murine peritonitis. Graphs show: (**A**) Representative dot plots following flow cytometry analysis indicating neutrophils as Ly6G+/CD11b+ double positive cells (upper right quadrant) in mice treated with LPS alone (0.1 mg/kg, 100 µL i.p.) or LPS+Erucin (1 µmol/kg, 100 µL i.p.) compared to control animals (Veh). (**B**) Scattered dot plot graph indicating % of Ly6G+/CD11b+ double positive cells peritoneal lavage following flow cytometry staining (n = 5). Results are shown as mean ± SEM. ** *p* < 0.01 vs. Veh. § *p* < 0.05 vs. LPS alone.

**Figure 2 ijms-23-15593-f002:**
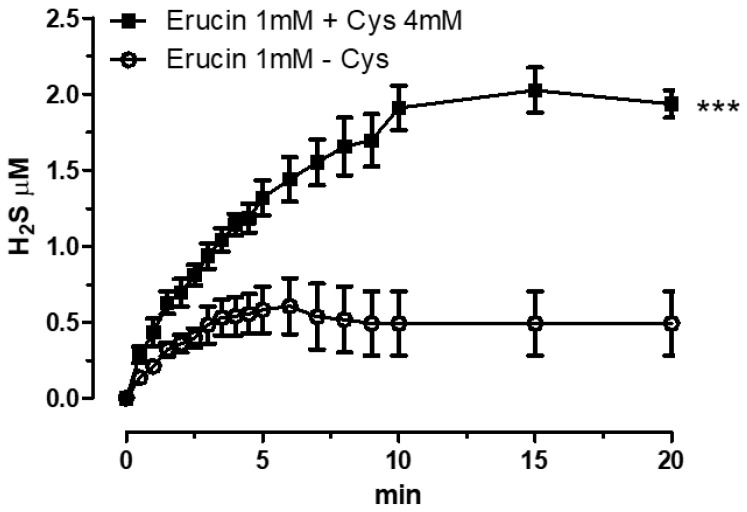
Amperometric recording of the H_2_S-releasing properties of erucin. The lines represent the increase in H_2_S during time (20 min) released by erucin in the presence and in the absence of L-Cysteine (Cys 4 mM). Results are shown as mean ± SEM. *** *p* < 0.001 vs. H_2_S release in the absence of L-Cysteine.

**Figure 3 ijms-23-15593-f003:**
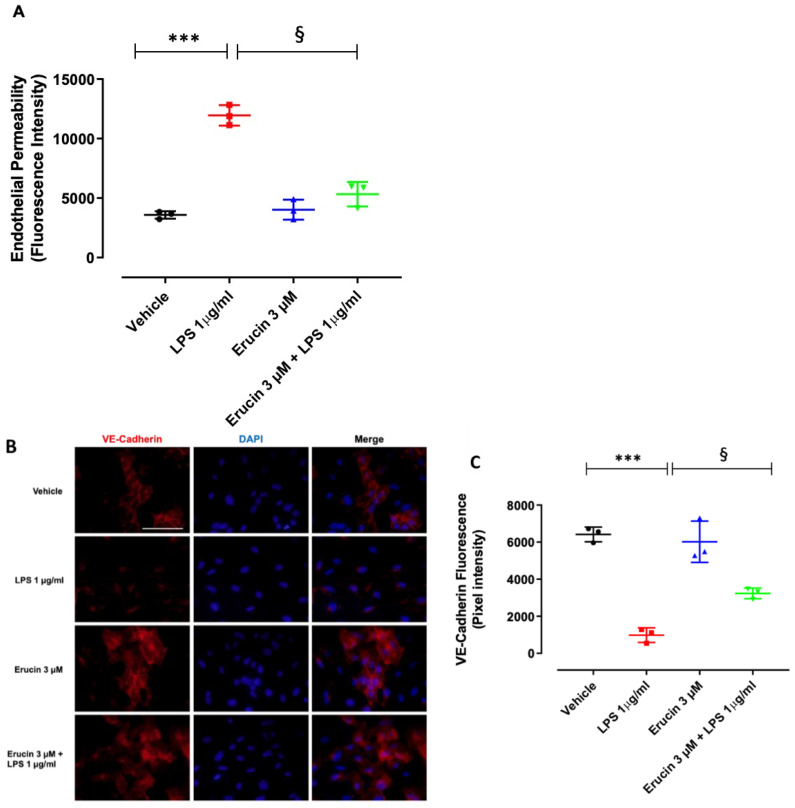
Effect of erucin on endothelial hyperpermeability induced by LPS. (**A**) Paracellular flux on HUVECs pre-treated with erucin (3 µM, 1 h) and then with LPS (1 µg/mL, 1 h). FITC-dextran transport in the lower compartment was measured at the end of LPS stimulation. (**B**) Integrity of cell–cell contacts evaluated by immunofluorescence analysis of VE-Cadherin in HUVEC monolayers pre-treated with erucin (3 µM, 1 h) and then with LPS (1 µg/mL, 1 h). Images were obtained by confocal microscope. Scale bar—50 µm. (**C**) Fluorescence intensity was measured on four images per slide by using Fiji software. Results are shown as mean ± SD. *** *p* < 0.001 vs. vehicle; § *p* < 0.05 vs. LPS 1 µg/mL.

**Figure 4 ijms-23-15593-f004:**
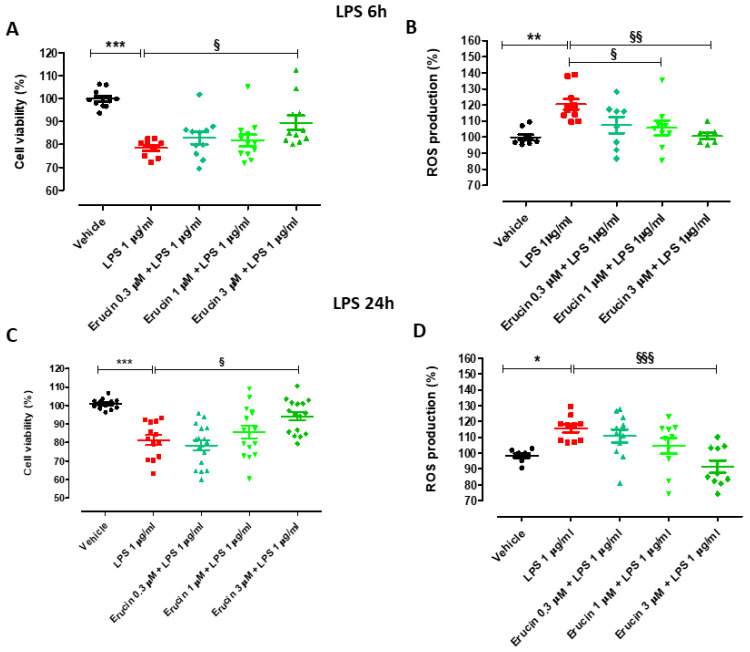
Effects of erucin on cell viability and ROS production in HUVECs treated with LPS (1 µg/mL). Graphs show cell viability (**A**) and intracellular ROS levels (**B**) of HUVECs treated with erucin (0.3, 1, and 3 µM) or vehicle (0.03% DMSO) for 1 h and then, with LPS 1 µg/mL co-incubated for 6 h. Graphs (**C**,**D**) show cell viability and intracellular ROS production after 24 h of treatment with LPS in the same experimental conditions as above. Results are shown as mean ± SEM. * indicates significant difference vs. vehicle (* *p*<0.05; ** *p* < 0.01; *** *p* < 0.001), and § indicates significant difference vs. LPS 1 µg/mL (§ *p* < 0.05; §§ *p* < 0.01; §§§ *p*<0.001).

**Figure 5 ijms-23-15593-f005:**
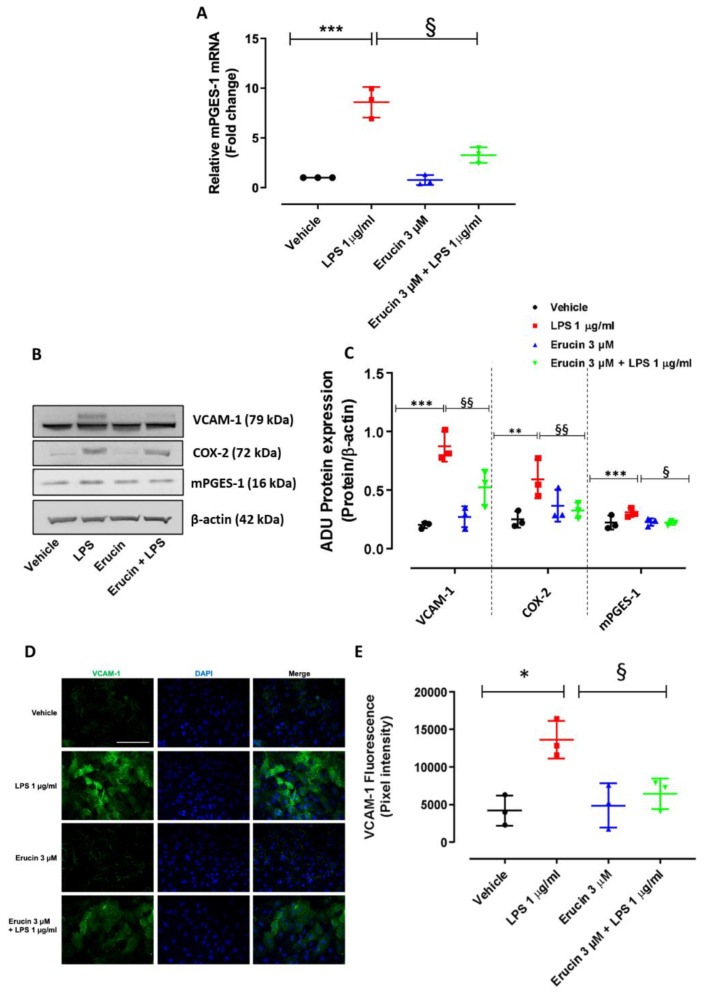
Effects of erucin on inflammatory markers and enzymes in endothelial cells treated with LPS. (**A**) mPGES-1 gene expression evaluated by RT-PCR. (**B**) Western blot analysis of inflammatory markers: VCAM-1, COX-2 and mPGES-1. Each lane contains 50 µg of total proteins obtained from endothelial cells pre-treated with erucin (3 µM, 1 h) and then exposed to LPS (1 µg/mL, 6 h). Blots are representative of three different experiments. (**C**) Quantification of the optical density of each protein of interest respect to its β-actin (n = 33). (**D**) VCAM-1 localization in HUVECs evaluated by immunofluorescence. Cells were pre-treated with erucin (3 µM, 1 h) and then with LPS (1 µg/mL, 6 h). Images were obtained by confocal microscope. Scale bar—50 µm. (**E**) VCAM-1 fluorescence intensity was measured on four images per slide by using Fiji software. Results are shown as mean ± SD. * *p*<0.05, ** *p* < 0.01 and *** *p* < 0.001 vs. vehicle; § *p* < 0.05 and §§ *p* < 0.01 vs. LPS 1 µg/mL.

**Figure 6 ijms-23-15593-f006:**
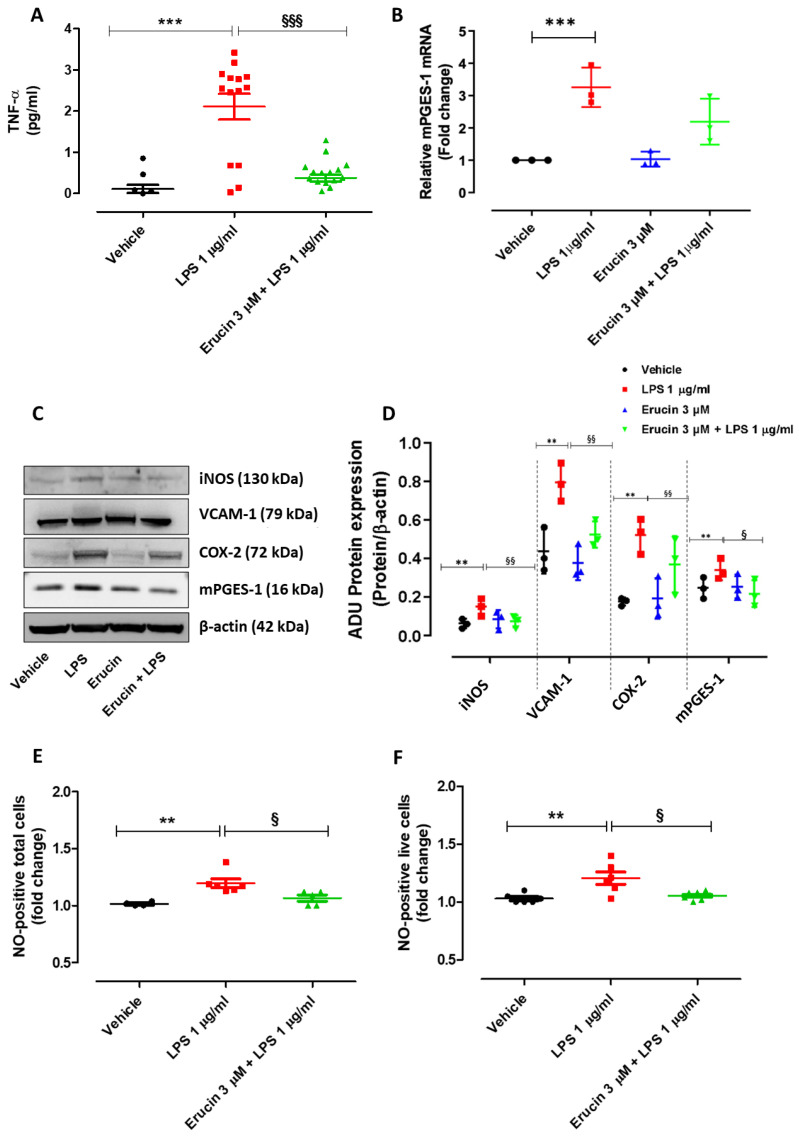
Preventive effects of erucin against inflammatory markers upregulation. TNF-α production (**A**) in HUVECs treated with erucin (3 µM) or vehicle (0.03% DMSO) for 1 h, and then with LPS 1 µg/mL co-incubated for 24 h. Graph shows the levels of TNF-α (pg/mL) measured in cell supernatants. Results are shown as mean ± SEM. (**B**) mPGES-1 gene expression evaluated by RT-PCR. (**C**) Western blot analysis of VCAM-1 and inflammatory markers: iNOS, COX-2 and mPGES-1. Each lane contains 50 µg of total proteins obtained from endothelial cells pre-treated with erucin (3 µM, 1 h) and then exposed to LPS (1 µg/mL, 24 h). (**D**) The graph represents the quantification of the optical density of each protein of interest with respect to its β-actin. Results are shown as mean ± SD (n = 33). Graph (**E**) shows the change in NO-positive total cells (live and dead), while (**F**) shows the change in NO-positive live cells treated with erucin (3 µM) or vehicle (0.03% DMSO) for 1 h, and then with LPS 1 µg/mL co-incubated for 24 h. Results are shown as mean ± SEM. * indicates significant difference vs. vehicle (** *p* < 0.01; *** *p* < 0.001), and § indicates significant difference vs. LPS 1 µg/mL (§ *p*<0.05; §§ *p* < 0.01; §§§ *p* < 0.001).

**Figure 7 ijms-23-15593-f007:**
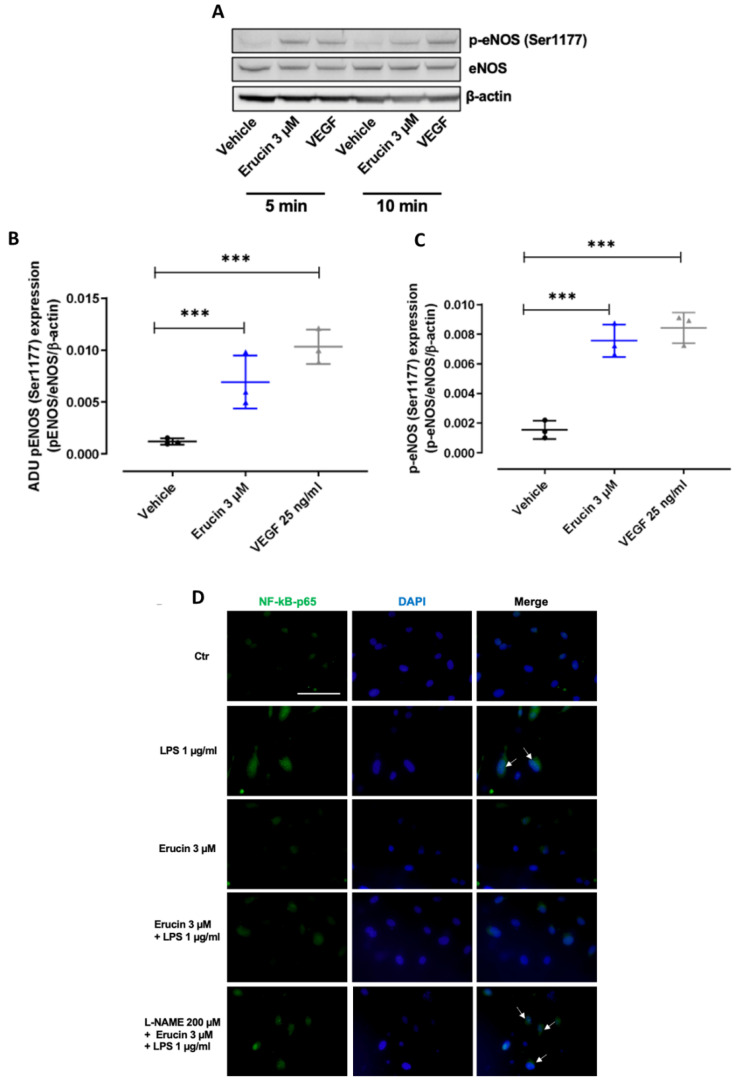
Molecular mechanism elicited by erucin in HUVECs. (**A**) eNOS activation on Ser1177 after 5 and 10 min of erucin treatment (3 µM). VEGF (25 ng/mL) was used as positive control of eNOS fast activation. The ratio between peNOS Ser1177 and eNOS is reported. Quantification of the optical density of each protein of interest with respect to total eNOS and its β-actin at 5 min (**B**) and 10 min (**C**). (**D**) NF-κB p65 subunit localization (in green, magnification 63×). HUVECs were starved for 4 h, and then pre-treated with erucin (3 µM) for 1 h followed by LPS (1 µg/mL, 2 h). Where indicated, HUVECs were pre-incubated with L-NAME (200 µM, 30 min). White arrows evidence nuclear NF-κB positivity. *** *p* < 0.001 vs. vehicle.

## Data Availability

The data presented in this study are available on request from the corresponding author.

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
