# Peer review of "Anti-Inflammatory Effect of the Natural H2S-Donor Erucin in Vascular Endothelium"

_ijms, 2022, doi:10.3390/ijms232415593_

Round 1

Reviewer 1 Report (New Reviewer)

The present study investigated the antioxidant and anti-inflammatory properties of H2S-donor erucin both in vivo and in vitro. Although the study is rich in data, the conceptual novelty is limited. This reviewer has the following concerns and suggestions:

Major

  1.  The antioxidant and anti-inflammatory effects of erucin have been studied in the authors’ previous study (PMID: 34203803). They have shown that erucin decreased the levels of inflammatory markers and endothelial hyperpermeability in an inflammatory model induced by high glucose. In this study, the authors just changed the stimuli to LPS and concluded the same conclusion without any mechanistic interrogation.
  2. In Figure 5B, 5C and 7C, it seems that the protein levels of VCAM-1 increased slightly after LPS treatment. Please check the results and explain this. This reviewer strongly recommends that the authors provide all the original WB results to prove the accuracy of the results.
  3. In Figure 5D and 5E, the statistical result was not consistent with the immunofluorescence.
  4. The activity of eNOS is canonically regulated by NF-kB. The authors used L-NAME to block the production of NO and then found that the inhibitory effect of erucin on LPS-induced NF-kB p65 subunit nuclear localization can be partially abolished. The authors draw the conclusion that erucin promotes fast eNOS activation, which in turn decreases NF-kB p65 subunit nuclear translocation. The evidence is insufficient. This reviewer strongly recommends the authors perform the experiment of cytoplasmic and nuclear protein extraction and detect the protein level p65 in the nuclear.

 Minor

1. Please change the abbreviation of “V-CAM1” to “VCAM-1” . 

Author Response

Please see the attachment. As concerns  VCAM-1 blots requested, You can find the pdf file attached as "Non-published material" in this step of manuscript revision.

Reviewer 2 Report (New Reviewer)

The reviewed manuscript entitled ‘Anti-inflammatory effect of the natural H2S-donor erucin in vascular endothelium’ written by Valerio Ciccone et al. presents very interesting results concerning the preventive properties of natural compound erucin against endothelial inflammation. The authors investigated potential mechanisms involved in this effect using various approaches. The article is well structured and scientifically sound. I have only minor comments on this manuscript.

11. The last two sentences of the introduction section: ‘Indeed, LPS is a component of outer envelope of Gram-negative bacteria. After infection, LPS is released from outer envelope and initiates a series of inflammatory responses, often leading to damaging of vascular endothelial cells.’ seems to be unnecessary and could be removed.

22. Paragraphs 2.4 and 2.6 in the Results section present results obtained by similar methods with the only difference with exposure time  (6h and 24h); therefore, it could be beneficial to merge these two paragraphs. It could make the text more clear.

33. In the western blot image provided in Figure 8A, the lanes for vehicle, erucin 3uM and VEGF are duplicated with no explanation why, please clarify this situation.

44. The presented work describes erucin as a promising substance with potential utility in the prevention of cardiovascular disease. However, it arises the important question of the safety of erucin administration. If the authors carried out the  safety and tolerability assessment of erucin in vivo, the results should be added to the manuscript. If not or if such an assessment is not possible to perform by the authors (e.g. due to required specific conditions of animal treatment), the results of earlier studies in this topic should be addressed in the Discussion section (of course, if such a results are available).

I believe that my suggestions will be helpful to the authors in increasing the quality of the reviewed manuscript.

Round 2

Reviewer 1 Report (New Reviewer)

The authors have well addressed the comments.

This manuscript is a resubmission of an earlier submission. The following is a list of the peer review reports and author responses from that submission.

Round 1

Reviewer 1 Report

This manuscript tested the effect of the H2S donor erucin in endothelial cells stimulated with LPS. This is a continuation of previous work from the same group showing the antioxidant effect of erucin in endothelial cells and VSMCs. They demonstrated the protective effect of erucin was mediated by downregulation of NF-kB. Thus, the novelty of the present work is low.

·      Figure 2B, fluorescence signal for VE-Cadherin looks more nuclear/perinuclear than membrane cell contacts. Cells look out of focus. A better-quality image should be shown. As it is, there is no evidence of erucin increasing the localization of VE-Cadherin at cell-cell contacts.

·      In Figure 3A, the effect of erucin in cell viability is modest.

·      For Figure 4B, the lanes for VCAM and COX2 are different from the lanes for actin and MPEGS suggesting that the westerns are from different gels. The quality of the western blots should be improved.

·      Figure 5 does not add new information is a repetition of Figure 3 for a longer incubation time.

·      In Figure 6B, the expression of mPGES is similar in all lanes, which does not match the quantification in Figure 6C.

·      Data for the in vivo model is limited

Author Response

Answers to Reviewer 1:

This manuscript tested the effect of the H2S donor erucin in endothelial cells stimulated with LPS. This is a continuation of previous work from the same group showing the antioxidant effect of erucin in endothelial cells and VSMCs. They demonstrated the protective effect of erucin was mediated by downregulation of NF-kB. Thus, the novelty of the present work is low.

- Figure 2B, fluorescence signal for VE-Cadherin looks more nuclear/perinuclear than membrane cell contacts. Cells look out of focus. A better-quality image should be shown. As it is, there is no evidence of erucin increasing the localization of VE-Cadherin at cell-cell contacts.

We thank the referee for the comment. We have improved the quality of pictures of Figure 2B. Analysing multiple images, we confirm the cell-cell contact localization of VE-Cadherin following erucin treatment, as well as cytoplasmatic/perinuclear accumulation. In the revised version of manuscript we have indicated the VE-cadherin localization at cell membrane with white arrows and amended the results.

-In Figure 3A, the effect of erucin in cell viability is modest.

We agree with the reviewer 1 and we modified the “2.3” paragraph (section Results) to better describe the modest effect, according his/her suggestion.

-For Figure 4B, the lanes for VCAM and COX2 are different from the lanes for actin and MPEGS suggesting that the westerns are from different gels. The quality of the western blots should be improved.

The westerns of Figure 4B in the submitted version of our manuscript were corresponding to different runs (see uncropped gels). The quantifications were however made on each own beta-actin and for simplicity the blots were condensed in one single panel. A new experiment has been performed and the western blot reported in Figure 4B has a better quality. Quantification in panel C reports the means of all the experiments done (N=4)

-Figure 5 does not add new information is a repetition of Figure 3 for a longer incubation time.

We agree with the reviewer 1 and we changed the “2.5” paragraph (section Results) in order to better highlight the fact that a similar level of protection has been promoted by erucin 3microM, both after 6h of incubation and after a prolonged exposure (24h) to LPS.

-In Figure 6B, the expression of mPGES is similar in all lanes, which does not match the quantification in Figure 6C.

We agree with reviewer. A new experiment has been performed and the statistical analysis of (n=4) experiments reveals that no statistical significance occurred.  The results have been changed accordingly.

-Data for the in vivo model is limited

We definitely agree with the reviewer 1 about the fact that the in vivo data presented in our work are limited to a single model of inflammation. In our view, the LPS induced peritonitis served as proof of the in vivo antinflammatory effect of erucin that has been described in all data set in vitro. The antinflammatory effect of erucin on leukocytes transmigration is consistent with that observed in cells with respect to adhesion molecule expression and vascular permeability. Of course, more models could have been used for better dissect the action triggered by erucin at interface between endothelial cells and leukocytes; however this was not our primary scope and might represent a follow up approach to investigate the behavior of erucin in vivo as a possible donor to modulate the biology of inflammation.

Moreover, because of the reviewer 1 suggested “moderate English changes”, we carefully edited the English language (you can see the edited text in the “Track Changes” function).

Reviewer 2 Report

I find the present article to be very important in the scientific field. It is very well written. The work performed on erucin was also very complex and the results are valuable for the cardiovascular field.

I would suggest adding an Abbreviation List at the end of the manuscript.

Moreover, in the discussion part please add some limitations of the study and future prespectives.

Author Response

Answers to Reviewer 2:

I find the present article to be very important in the scientific field. It is very well written. The work performed on erucin was also very complex and the results are valuable for the cardiovascular field.

We thank the reviewer 2 very much for is/her appreciation

I would suggest adding an Abbreviation List at the end of the manuscript.

We agree with the reviewer 2 and, as a consequence of his/her suggestion, we added a list of abbreviation at the end of the manuscript.

Moreover, in the discussion part please add some limitations of the study and future prespectives.

We thank the reviewer 2 for his/her suggestion that improve our manuscript and we added two paragraphs about limitations and future perspectives in the Discussion part, as suggested.

Round 2

Reviewer 1 Report

The response of authors to previous comments were not satisfactory.

1.     Authors did not comment on the low novelty of their work.

2.     Images for Fig. 2B are the same as before, they only added arrows to the images.

3.     For Fig. 4B, authors explain that they ran new western blots, but the image for mPEGEs is the same as before.